# Approximate Solutions of Nonlinear Partial Differential Equations Using B-Polynomial Bases

**Muhammad I. Bhatti \*, Md. Habibur Rahman** and **Nicholas Dimakis**

Department of Physics and Astronomy, University of Texas, Rio Grande Valley, Edinburg, TX 78539, USA;
mdhabibur.rahman02@utrgv.edu (M.H.R.); nicholas.dimakis@utrgv.edu (N.D.)
\* Correspondence: muhammad.bhatti@utrgv.edu

**Abstract:** A multivariable technique has been incorporated for guesstimating solutions of Nonlinear Partial Differential Equations (NPDE) using bases set of B-Polynomials (B-polys). To approximate the anticipated solution of the NPD equation, a linear product of variable coefficients $a_i(t)$ and $B_i(x)$ B-polys has been employed. Additionally, the variable quantities in the anticipated solution are determined using the Galerkin method for minimizing errors. Before the minimization process is to take place, the NPDE is converted into an operational matrix equation which, when inverted, yields values of the undefined coefficients in the expected solution. The nonlinear terms of the NPDE are combined in the operational matrix equation using the initial guess and iterated until converged values of coefficients are obtained. A valid converged solution of NPDE is established when an appropriate degree of B-poly basis is employed, and the initial conditions are imposed on the operational matrix before the inverse is invoked. However, the accuracy of the solution depends on the number of B-polys of a certain degree expressed in multidimensional variables. Four examples of NPDE have been worked out to show the efficacy and accuracy of the two-dimensional B-poly technique. The estimated solutions of the examples are compared with the known exact solutions and an excellent agreement is found between them. In calculating the solutions of the NPD equations, the currently employed technique provides a higher-order precision compared to the finite difference method. The present technique could be readily extended to solving complex partial differential equations in multivariable problems.

**Keywords:** multivariable linear and nonlinear differential equations; multidimensional formulation; B-polynomial basis set; hyperbolic partial differential equations

## 1. Introduction

Using a B-polynomial basis set, one can solve very complicated 2D partial differential equations which could appear in the fields of physics, engineering, chemistry, and computer science [1,2]. The flawless integration and differentiation are the nature of B-polys that aid in the use of symbolic programming languages, such as Mathematica or Maple. Over any closed interval, B-polys are smooth functions that provide the basis to represent an arbitrary function to desirable correctness [1–3]. The specific details and properties of the B-polys are provided in our previous works [1–3]. Briefly, B-polys of nth degree are defined as $B_{i,n}(x) = \begin{pmatrix} n \\ i \end{pmatrix} \frac{(x-a)^i (b-x)^{n-i}}{(b-a)^n}$, for $i = 0, 1, 2 \ldots n$, where binomial coefficients are given by $\begin{pmatrix} n \\ i \end{pmatrix} = \frac{n!}{n!(n-i)!}$. The $(n + 1)$ B-polys form a complete basis set over the interval $[a, b]$.

In the earlier years, various methods were employed to solve linear and nonlinear differential equations, including fractional-order differential equations [4–13]. In the articles [1–3], the authors have solved various differential equations employing a concoction of operational matrix and B-poly basis set. In the earlier progression, the authors used

B-poly technique to calculate solutions of the single-variable differential equations [1]. In our recently extended task, two-variable dependent Hyperbolic Partial Differential (HPD) equations have been solved using the B-poly bases [13]. In the year 2011, the B-polys were expressed in terms of Legendre basis that has been used to solve the linear differential equations [14]. The authors, Bhatti and Hinojosa, have successfully applied similar progression to the linear partial differential equations and highly accurate solutions were reported [13]. It is observed that the B-poly method provides better accuracy than finite difference method. For example, Kutluay et al. [15] have presented a finite difference solution to the Burgers' equation. It is shown that the accuracy of the solution depends on $h^2$ in the finite difference method, where h is the step size. This requires a very large number of grid points to accurately represent the solution in that region. In reference, Bhatti and Bhatta [16], a combination of B-poly and fourth order Runge–Kutta method was applied to show that B-poly technique provided a higher order of accuracy (four orders of magnitude better) in the solution of the Burger equation.

In the present work, our goal is to apply an extended version of the technique for solving two variables (*x*,*t*) NPD equations by means of B-poly basis, operational matrix, and the Galerkin method [17]. It is well known that the continuous and unitary property of B-poly help to determine semi-analytic and, in some cases, exact solutions in a much quicker way in terms of CPU time [1]. The newly designed progression has been effectively used for solving two variables NPD equations in closed intervals, such as [0,R] and [0,T]. A complete basis set of polynomials in two variables (*x*,*t*) in terms of the product of B-poly sets has been utilized to figure out the solution of the NPDE. Two types of general second-order nonlinear partial differential equations are considered in this paper. These equations are used for modeling real-world physical phenomena in the fields of physics and mathematics. The equations have applications in various fields of science, like fluid dynamics, heat conduction, elasticity, traffic control simulations, and number theory [16], also see references therein. Further, these equations can be converted into well-known Burgers' equation if the right-hand side is set equal to zero and proper values of constants are chosen. The importance of these calculations is that sometimes they present exact solutions which can be used to judge algorithms as well as the accuracy of computational techniques [13,16]. These equations have nonlinear convective terms and second-order diffusion terms which act opposite to each other. Furthermore, these terms allow to admit distorted wave profile as a solution which is difficult to solve numerically.

The technique is explained step by step and applied to a more general two-dimensional NPDE,

$$\alpha \frac{d^2 y(x,t)}{dx^2} + \beta\, y(x,t) \frac{dy(x,t)}{dx} + \gamma\, \frac{dy(x,t)}{dt} = g(x,t). \tag{1}$$

where $\alpha$, $\beta$, and $\gamma$ could be constant or variables. A desired solution of the NPDE is expressed as a linear combination of B-poly basis set, as follows.

$$y(\mathrm{x},\mathrm{t}) = \sum_{i=0}^{n} a_i(t)\, B_{i,n}(x), \tag{2}$$

where $a_i(t)$ is the *i*-th expansion unknown coefficient in Equation (2) that is a function of variable *t*. In Equation (2), we impose the initial conditions on variables (*x*,*t*). The $B_{i,n}(x)$ is the *n*-th degree B-poly in variable *x* from the basis set. Furthermore, the coefficients $a_i(t)$ could be expressed in terms of the constant coefficients $b_j^i \cdot$ and the B-polys $B_{j,m}(t)$ in variable *t* that could have same or a different set of polynomials over the interval [0,T], such as,

$$a_i(t) = \sum_{j=0}^{m} b_j^i\, B_{j,m}(t). \tag{3}$$

In Equation (3), the coefficients $a_i(t)$ could be subjected to initial and boundary conditions if required. We plan to present results by using a complete set of B-polys of the different degree to a few nonlinear partial differential equations. In the following

sections, the procedure is applied to solve equations, and the results of the NPD equations are compared with the available 2D exact solutions. An excellent agreement has been found between the exact and estimated solutions. In the previous work, following the notation in [1–3,13], the work has been protracted to include complete B-poly bases sets that were involved to approximate results of a variety of differential equations [13,16].

The current technique is applied by substituting the approximate solution, Equation (2), into the NPDE (1) to separate out inner products in variables $x$ and $t$. Both sides of the equations are multiplied by a product of B-polys, $B_m(x) \, B_n(t)$, and integration over the closed intervals is carried out. The inner products of B-polys are multiplied to form an operational matrix with a nonzero determinant. Finally, the inverse of the operational matrix is carried out to determine the unknown coefficients of the linear combination. The coveted solution of the NPDE is assembled with the initial condition imposed on the operational matrix equation. In the following sections, we shall explain the process of how to find an appropriate solution, present plots of the solutions, and calculate semi analytic solutions for each of the four examples considered in this paper. Comparisons between exact and approximate solutions will be made in Section 2 and, finally, error analysis of the final example will be presented in Section 3.

## 2. Computations of Solutions of NPD Equations

As mentioned in the previous section, details of the B-polys will be left out to avoid duplication of the formulas. Furthermore, details on how to generate sets of B-poly basis and construct a solution from the sets are provided [1,2,13]. To make things simpler, once the degree of B-poly basis set is chosen, we may neglect subscript $n$ representing the degree in B-polys $B_{i,n}(x)$. The current technique to estimate solution, $y(x, t)$, of the two-dimensional differential Equation (1) is outlined in this section. The solution is considered as a combination of the variables, $a_i(t) \ and \ B_i(x)$. The NPDE is solved by imposing an initial condition on Equation (3), employing the Galerkin method [17], and taking the inverse of the operational matrix for calculating the unknown variables, $a_i(t)$ [1,13]. The approximate solution with the initial condition $y(x, \, 0) = f(x)$ is given by [13],

$$y(x, t) = \sum_{i=0}^{n} a_i(t) \, B_i(x) + f(x). \tag{4}$$

Implementing this presumed solution, Equation (4), into a nonlinear partial differential Equation (1), we may transform NPDE into an operational matrix by calculating the inner products of B-poly in both variables $x$ and $t$. The initial condition at $t = 0, f(x) = y(x,0)$ is added to the solution given in Equation (4). The solution is substituted into the differential Equation to convert it into operational matrix $X$ which is inverted to find the solution [16]. Before we find its inverse to be applied for determining unknown coefficients, boundary/initial conditions are applied to the operational matrix and the right-hand side matrix in the interval $[a, b]$. This is done by deleting the first row and the first column of the operational matrix and the corresponding entry of the column matrix to ensure solution vanishes at the origin. Below, we have provided four examples of the NPD equations which are solved using the proposed technique and the initial condition $y(x,0) = f(x)$ at $t = 0$. Putting Equation (4) into the second order NPD Equation (1), we have equation,

$$\alpha \sum_{i=0}^{n} a_i(t) B_i''(x) + \alpha f''(x) \quad + \beta \left( \sum_{j}^{n} a_j(t) B_j(x) + f(x) \right) \left( \sum_{i}^{n} a_i(t) B_i'(x) + f'(x) \right) + \gamma \sum_{i}^{n} \dot{a}_i(t) \, B_i(x) \\ = g(x,t). \tag{5}$$

where dot ($\cdot$) and prime ($'$) denote derivatives with respect to $t$ and $x$, respectively. The Equation (5) can be further simplified by moving some of the terms that do not depend on unknown coefficients $a_i(t)$ to the right-hand side of the equation,

$$\sum_{i=0}^{n} a_i(t) \left[ \ \alpha B_i''(x) \quad + \beta B_i'(x) \sum_{j=0}^{n} a_j(t) \ B_j(x) + \beta f(x) B_i'(x) + \beta f'(x) B_i(x) \ \right] + \gamma \sum_{i=0}^{n} \dot{a}_i(t) \ B_i(x) \tag{6}$$
$$= g(x,t) - \beta \ f'(x) f(x) - \alpha f''(x).$$

The expansion of $a_i(t) = \sum_{j=0}^{n} b_j^i B_j(t)$ and $a_k(t) = \sum_{l=0}^{n} b_l^k B_l(t)$ coefficients can be used to convert Equation (6) in terms of constants, $b_j^i$. After multiplying both sides of the Equation (6) with the product of B-polys $B_m(x) \ B_n(t)$ on both sides and integrating with respect to $t$ and $x$ over the intervals $t \in [0, \ T]$ and $x \in [0, \ R]$, the Equation (6) is transformed into Equation (7),

$$\sum_{j=0}^{n} \sum_{i=0}^{n} b_j^i [\langle \alpha B_i''(x) + \beta f(x) \ B_i'(x) + \beta f'(x) \ B_i(x) | B_m(x) \rangle \langle B_j(t) | B_n(t) \rangle$$

$$+ \beta \sum_{k,l}^{n} b_l^k g_{ikm}' \ g_{jln} + \gamma \langle B_i(x) | B_m(x) \rangle \langle \dot{B}_j(t) | B_n(t) \rangle]$$

$$= \langle \langle g(x,t) - \beta \ f'(x) f(x) - \alpha f''(x) | B_m(x) \rangle | B_n(t) \rangle.$$

The integrals and inner products of B-polys including the nonlinear terms are given below:

$$\langle B_m(x) | B_n(x) \rangle = \int_0^R B_m(x) \ B_n(x) \ dx$$

$$\langle B_m(t) | B_n(t) \rangle = \int_0^T B_m(t) \ B_n(t) \ dt$$

$$\langle \langle B_i(x) | B_m(x) \rangle | B_n(t) \rangle = \iint\limits_0^{R,T} B_i(x) B_m(x) B_n(t) dx \ dt.$$

$$g_{ikm}' = \int_0^R B_i'(x) \ B_k(x) B_m(x) dx$$

$$g_{jln} = \int_0^T B_j(t) B_l(t) B_n(t) dt$$
$$\tag{7}$$

The terms $g_{ikm}$ & $\dot{g}_{jln}$ are factors that exhibit the nonlinear terms due to the appearance of the coefficients $b_l^k$ in front of them. Using the initial condition and including the nonlinear terms, we can carry out calculations of the coefficients $b_l^k$. The converged coefficients are determined after a few iterations of the nonlinear terms in the above Equation (7). Sometimes, the initial guess to the nonlinear terms may be given zero in the first iteration. We also employ the Galerkin method [17] to minimize the error in the solution of the nonlinear partial differential equations. In this method, the coefficients in Equation (4) are minimized by increasing or decreasing a certain degree and number of polynomials in the process.

We may consider another type of general nonlinear PDE of the form in which nonlinear term appears at a different place in the equation,

$$\alpha \frac{d^2 y(x,t)}{dx^2} + \beta \frac{dy(x,t)}{dx} + \gamma \ y(x,t) \frac{dy(x,t)}{dt} = g(x,t). \tag{8}$$

Notice that now the nonlinear term appears in the third term as opposed to the second term in Equation (1). Using similar steps as described above, we can change the above Equation (8) into an operational matrix,

$$\sum_{j=0}^{n} \sum_{i=0}^{n} b_j^i \left[ \alpha B_i''(x) + \ \beta B_i'(x) | B_m(x) B_j(t) | B_n(t) + \gamma f(x) B_i(x) | B_m(x) \dot{B}_j(t) | B_n(t) + \gamma \sum_{n}^{k,l=0} b_l^k g_{ikm} \ \dot{g}_{jln} \right] \tag{9}$$
$$= \langle \langle g(x,t) - \beta \ f'(x) - \alpha f''(x) | B_m(x) \rangle | B_n(t) \rangle.$$

where, $\dot{g}_{jln}$ & $g_{ikm}$ are the nonlinear terms which exhibit nonlinearity via the coefficients $b_l^k$. The results of the Equation (9) are further updated at each iteration when it is subjected to the initial condition and initial guess for the nonlinear term. The error in the solution of NPDE is minimized using the Galerkin method [17]. The current technique is applied to

four nonlinear differential examples. It is demonstrated that the technique is suitable to find an accurate solution to NPDE.

**Example 1.** *Consider a NPDE which is obtained presuming parameters* $\alpha = 0$, $\beta = t$, $\gamma = -1$ *and* $g(x, t) = x$ *in the general Equation (8). The NPDE in two dimensions is given,*

$$t\frac{dy}{dx} - y\frac{dy}{dt} = x. \tag{10}$$

The exact solution of the Equation (10) is $y_{exact}(x, t) = (x - t)$. For evaluating numerical solution using initial condition $y(x, 0) = f(x) = x$ in intervals $0 \leq t \leq 1$ and $0 \leq x \leq 1$, an approximate solution to Equation (10) may be assumed as $y(x, t) = \sum_{i=0}^{n} a_i(t) B_i(x) + x$. After substituting this expansion into Equation (10), we get a similar equation as Equation (9) which restructures to the following equation,

$$\sum_{i,j=0}^{n} b_j^i \left[ \langle B_i'(x)|B_m(x)\rangle\langle t\, B_j(t)|B_n(t)\rangle - \langle x\, B_i(x)|B_m(x)\rangle\langle \dot{B}_j(t)|B_n(t)\rangle \right.$$
$$\left. - \sum_{k,l=0}^{n} b_l^k g_{ikm}\, \dot{g}_{jln} \right] = \langle\langle x - t|B_m(x)\rangle|B_n(t)\rangle. \tag{11}$$

where, $g_{ikm}$ & $\dot{g}_{jln}$ are the nonlinear terms which exhibit nonlinearity via the unknown coefficients $b_l^k$. This algorithm leads to an $(n + 1)$ by $(n + 1)$ system of equations $X B = W$, in unknown variables $B = \{b_1^1, b_2^1, b_3^1, \ldots, b_1^2, b_2^2, b_3^2, \ldots, \}$, which are elements of matrix $B$. The matrices $X$ *and* $W$ in terms of inner products of B-polys are provided below,

$$X_{m,n} = \sum_{i,j=0}^{n} \left[ \langle B_i'(x)|B_m(x)\rangle\langle t\, B_j(t)|B_n(t)\rangle - \langle x\, B_i(x)|B_m(x)\rangle\langle \dot{B}_j(t)|B_n(t)\rangle - \sum_{k,l=0}^{n} b_l^k g_{ikm}\, \dot{g}_{jln} \right],$$
$$W_{m,n} = \langle\langle x - t|B_m(x)\rangle|B_n(t)\rangle = \iint_0^{R,T}(x - t)B_m(x)B_n(t)dx\, dt. \tag{12}$$

The nonlinear partial differential Equation (12) has been converted into an operational matrix form $X$ given in Equation (12), whose inverse is multiplied by the column matrix W to yield values of the unknown coefficients $b_j^i$ by solving the equation $B = X^{-1}W$. Before constructing the operational matrix equation, $X B = W$, initial conditions are imposed by deleting rows and the corresponding columns of the Equation (12), so that the solution vanishes at $t = 0$ and $x = 0$. The resulting approximate solution is gathered from the B-poly basis set via Equations (3) and (4). After two iterations the converged values of the coefficients $b_j^i$ were attained, $\{0, -1, 0, -1\}$. With the application of this technique, the solution of Equation (10) is obtained, which is given for intervals $t \in [0, 1]$ and $x \in [0, 1]$,

$$y(x, t) = t\left(-1.0 + 0. \times 10^{-30}x\right) + x \approx x - t. \tag{13}$$

As you may note, the above result is very accurate. To solve the NPD Equation (10), the B-polys of degree n=1 have been utilized in both variables $x$ and $t$. The B-poly basis sets used are $\{1 - t, t\}$ and $\{1 - x, x\}$ which gave a $4 \times 4$ operational matrix by multiplying both sets. We have presented 3D plots of the exact and estimated results of Equation (13) for comparison, see Figure 1, which shows exact agreement between both solutions at the level of machine precision. Note that when $t = x$ is replaced in the Equation (13), the error can be seen of the order $10^{-17}$ representing the high quality of the resolution in one-dimension $x$. The solution is essentially zero when $t = x$ is replaced. In this example, the absolute error between the solutions is negligible, showing that both solutions are in perfect agreement.

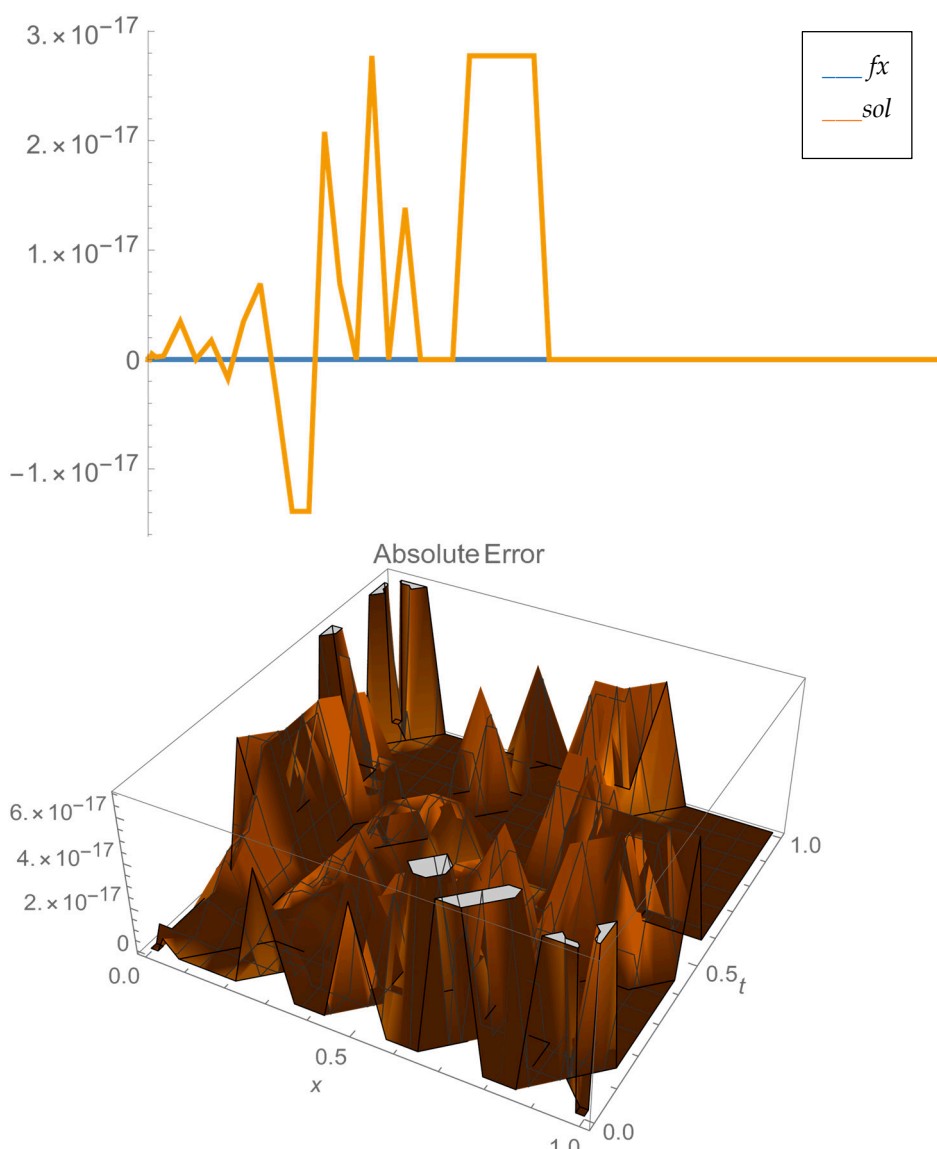

**Figure 1.** The illustrations of the contrast between exact (*sol*) and approximate (*fx*) solutions are presented on the left-hand for $t = x$ replaced in the solutions of Equation (10). This shows complete overlap of the solutions. On the right-hand side, a 3D plot of the absolute error between exact and estimated solutions is also shown in the intervals $x \in [0, 1]$ and $t \in [0, 1]$. The graph shows the accuracy of the numerical results is of the order of $10^{-17}$. This kind of accuracy occurred with an only $n = 1$-degree polynomial basis set.

**Example 2.** *Another case of the NPDE with a distinct nonlinear term is considered. We replace $\alpha = 0$, $\beta = 1$, $\gamma = 1$, and $g(x, t) = x - t - 1$ in the general NPD Equation (1). This equation has a nonlinear term associated with the first term as opposed to the nonlinearity term associated with the second term of the first example. We want to show that the current technique can handle nonlinearity terms associated to any term in the NPDE. The second example adds another level of difficulty to be considered with nonhomogeneous terms on the right-hand side of Equation (14). The NPDE in two dimensions is given by,*

$$y \frac{dy}{dx} + \frac{dy}{dt} = x - t - 1. \tag{14}$$

We are searching for a solution of Equation (14) in the intervals $0 \leq t \leq 1$ and $0 \leq x \leq 1$ with initial condition $y(x, 0) = f(x) = x$ at $t = 0$. The exact solution to

Equation (14) is known, $y_{exact}(x, t) = (x - t)$. An approximate solution to Equation (14) may be assumed to be $y(x, t) = \sum_{i=0}^{n} a_i(t)\, B_i(x) + f(x)$. In the assumed solution $y(x, t)$, the coefficient $a_i(t)$ is the $i$-th coefficient in the expansion which depends on $t$. By replacing the assumed solution into Equation (14) and multiplying both sides of Equation (14) with the product of B-polys $B_m(x)\, B_n(t)$, we can separately perform integration over both variables $t$ and $x$ in the intervals $t \in [0, T]$ and $x \in [0, R]$, respectively. Applying an additional approximation to the coefficients $a_i(t) = \sum_{i=0}^{n} b_j^i B_j(t)$, the Equation (3) and with substitution of the term $g(x, t) = x - t - 1$, we attain Equation (15) given below:

$$\sum_{n}^{i,j=0} b_j^i \left[ \langle f(x) B_i'(x) + f'(x)\, B_i(x)|B_m(x) \rangle \langle B_j(t)|B_n(t) \rangle + \sum_{n}^{k,l=0} b_l^k\, g_{ikm}'g_{jln} + \langle B_i(x)|B_m(x) \rangle \langle \dot{B}_j(t)|B_n(t) \rangle \right]$$
$$= \langle \langle x - t - 1 - f(x)\, f'(x)|B_m(x) \rangle|B_n(t) \rangle. \tag{15}$$

where, $g_{ikm}'$ & $g_{jln}$ are the nonlinear terms linked via the unknown coefficients $b_l^k$. The Equation (15) leads to an $(n + 1)$ by $(n + 1)$ system of equations $X\,B = W$, in the unknown variables $B = \{b_1^1, b_2^1, b_3^1, \ldots, b_1^2, b_2^2, b_3^2, \ldots, \}$, elements of matrix $B$, where the matrices $X$ *and* $W$ are given,

$$X_{m,n} = \sum_{i,j=0}^{n} \left[ \langle f(x)B_i'(x) + f'(x)\, B_i(x)|B_m(x) \rangle \langle B_j(t)|B_n(t) \rangle + \sum_{k,l=0}^{n} b_l^k\, g_{ikm}'g_{jln} + \langle B_i(x)|B_m(x) \rangle \langle \dot{B}_j(t)|B_n(t) \rangle \right],$$
$$W_{m,n} = \langle \langle x - t - 1 - f(x)f'(x)|B_m(x) \rangle|B_n(t) \rangle. \tag{16}$$

The operational matrix $X$ is provided in Equation (16), whose inversion yields values of unknown coefficients $b_j^i$ through solving the equation $B = X^{-1}W$. Before the inverse of matrix $X$ is called for, initial conditions are imposed by deleting the appropriate rows and columns (deleted first row and first column so that the desired solutions vanish at the origin) of the matrices $X$ and $W$ to make the solution vanish at $t = 0$ and $x = 0$. Using a variational property with respect to the coefficients and carrying out several iterations of the nonlinear terms the converged solution of the Equation (14) is given,

$$y(x, t) = t\left(-1.0 + 0. \times 10^{-27}\right) + x \approx x - t. \tag{17}$$

The values of the coefficients are listed as $b_j^i = \{0, -1, 0, -1\}$ which are needed to construct the numerical solution via Equation (3). The B-poly basis sets used are $\{1 - t, t\}$ and $\{1 - x, x\}$ in both variables $(x, t)$, respectively. The expected solution, in the closed intervals, e.g., $t \in [0, 1]$ *and* $x \in [0, 1]$, has the accuracy of the order of $10^{-27}$. Again, only $n$ = 1-degree B-polys were required for solving the differential Equation (16) in both variables $(x, t)$. Figure 2 shows the level of agreement between both solutions at the machine precision. Note that when $t = x$ is replaced in the Equation (17), the error can be seen to be zero, providing the high quality of the resolution in one-dimension $x$ as well as in 3D.

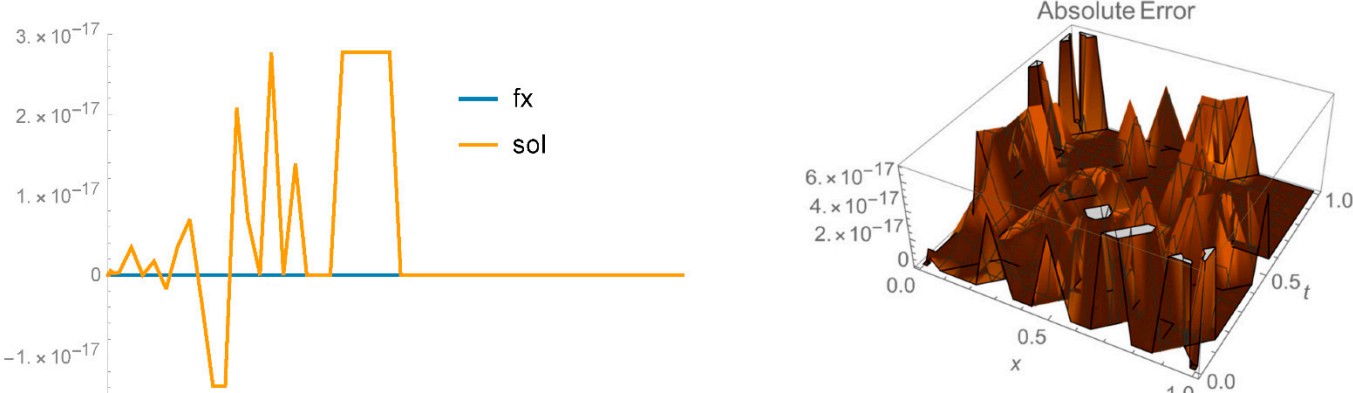

**Figure 2.** The illustrations of the contrast between precise (sol) and approximate (*fx*) results are presented when *t* = *x* is replaced in the solutions of the Equation (14), on the left-hand side of the figure. This graph shows accuracy in the single variable *x*. On the right-hand side of the figure, 3D plot of the absolute error between both solutions is also given over the intervals $x \in [0, 1]$ and $t \in [0, 1]$. The graphs show the high precision of the numerical results. This kind of accuracy occurred when n = 1-degree polynomials were used in variables (*x*, *t*) for approximating the solution of Equation (14).

**Example 3.** *Next, we consider a more complex second-order NPD equation with a distinct nonlinear term. By replacing parameters $\beta = -1$, $\alpha = \mu = 1$, $\gamma = x\,t^3$, and $g(x,t) = 2\,\mu\,t^2$ in the general NPD Equation (1), the NPD equation becomes,*

$$\mu \frac{d^2 y}{dx^2} - y\,\frac{dy}{dx} + x\,t^3 \frac{dy}{dt} = 2\,\mu\,t^2. \tag{18}$$

Again, we are searching for the solution in the intervals $0 \le x \le 1$ and $0 \le t \le 1$. Here we are going to use a slightly different initial condition, $y\,(1,t) = f\,(t) = t^2$. The desired solution for Equation (18) can be assumed, $y(x,t) = \sum\limits_{i=0}^{n} a_i(t)\,B_i(x) + t^2$. The unknown coefficient $a_i(t)$ is the *i*-th coefficient as a function of variable *t* in the evolution of expression, $y(x,t)$. By inserting this approximate solution into Equation (18) and multiplying both sides of Equation (18) with the product of B-polys $B_m(x)\,B_n(t)$, we perform integration separately over both variables x and t in the intervals $t \in [0,\ R]$ and $x \in [0,\ T]$, respectively. Additional approximation to the coefficients $a_i(t) = \sum\limits_{i=0}^{n} b_j^i B_j(t)$ was also used to obtain the following equation,

$$\sum_{i,j=0}^{n} b_j^i \big[\langle B_i''(x)|B_m(x)B_j(t)|B_n(t)\rangle - \langle B_i'(x)|B_m(x)\rangle\langle t^2 B_j(t)|B_n(t)\rangle$$

$$- \sum_{k,l=0}^{n} b_l^k\,g_{ikm}'g_{jln} + \langle x\,B_i(x)|B_m(x)\rangle\left\langle t^3 \dot B_j(t)|B_n(t)\right\rangle\big] = \langle\langle(2t^2 - 2\,x\,t^4)|B_m(x)\rangle|B_n(t)\rangle. \tag{19}$$

where, $g_{ikm}'$ & $g_{jln}$ are the nonlinear terms which exhibit nonlinearity via the unknown coefficients $b_l^k$. This algorithm leads to an $(n+1)$ by $(n+1)$ order of matrix equation $X\,B = W$, where unknown coefficients are $B = \{b_1^1, b_2^1, b_3^1, \ldots, b_1^2, b_2^2, b_3^2, \ldots, \}$, that represent elements of matrix *B*, and the matrices *X* and *W* are given,

$$X_{m,n} = \sum_{i,j=0}^{n} [\langle B_i''(x)|B_m(x)\rangle\langle B_j(t)|B_n(t)\rangle - \langle B_i'(x)|B_m(x)\rangle\langle t^2 B_j(t)|B_n(t)\rangle$$

$$- \sum_{k,l=0}^{n} b_l^k\,g_{ikm}'g_{jln} + \langle x\,B_i(x)|B_m(x)\rangle\langle t^3 \dot B_j(t)|B_n(t)\rangle], \tag{20}$$

$$W_{m,n} = \langle\langle(2t^2 - 2\,x\,t^4)|B_m(x)\rangle|B_n(t)\rangle = \iint\limits_{0}^{R,\ T}(2t^2 - 2\,x\,t^4)B_m(x)B_n(t)dx\,dt.$$

The NPDE is transformed into an operational matrix equation $X\,B = W$, where the inversion of the matrix $X$ yields values of the unknown coefficients $b_j^i$ in Equation (19) by solving equation $B = X^{-1}W$. The approximate result is constructed from the product of the B-poly basis set $B_i(t)$ and the coefficients $a_i(t)$. The converged values of the coefficients for this example are, $b_j^i = \{0,\ 0.\times 10^{-28}, 0.\times 10^{-27}, 0,\ 0.\times 10^{-28},\ 0.\times 10^{-27},\text{and } 0,\ 0.\times 10^{-27},\ 1.0\}$. The initial condition is imposed on the matrices X and W defined in Equation (20) by deleting the first row and the corresponding first column because the solution must be zero at $t = 0$ and $x = 0$. This approach delivered a valid solution of the Equation (18) and the estimated result is provided,

$$y(x,t) = t\left(0.\times 10^{-27} + 0.\times 10^{-27}x + 0.\times 10^{-27}x^2\right) + t^2\left(0.\times 10^{-27} + 0.\times 10^{-26}x + 1.0x^2\right) \approx x^2\,t^2. \tag{21}$$

The numerical result in Equation (21) is compared with the exact solution $y_{exact}(x,t) = x^2\,t^2$. Obviously, the approximate solution has high precision in the closed intervals $t \in [0,\,1]$ and $x \in [0,\,1]$ that matched the exact solution when contributions of the small terms of the order $\sim 10^{-27}$ were discounted. This kind of accuracy shown in Equation (21) was achieved with $n = 2$-degree B-polys in both variables $(x,t)$ which gave a $9 \times 9$ dimension operational matrix. To obtain such a high degree accuracy of the solution, we used three B-polynomials in $x$-variable $\{1 - 2\,x + x^2,\ 2\,x - 2\,x^2,\text{ and } x^2\}$ and a set of three B-polynomials in $t$-variable $\{1 - 2t + t^2,\ 2\,t - 2\,t^2,\text{ and } t^2\}$. To observe the accuracy level of the solution in only one variable $x$, we replaced $t = x$ in the approximate solution $(fx)$ and the exact solution $(sol)$ of Equation (18), we essentially found overlays of graphs shown on the left-hand side of Figure 3. We have also shown 3D graphs of both the exact and the estimated solutions on the right-hand side of Figure 3. It is observed that there is no appreciable difference between the graphs of both solutions as the error is very small and hence indicating the technique works for calculating the solution of the NPDE.

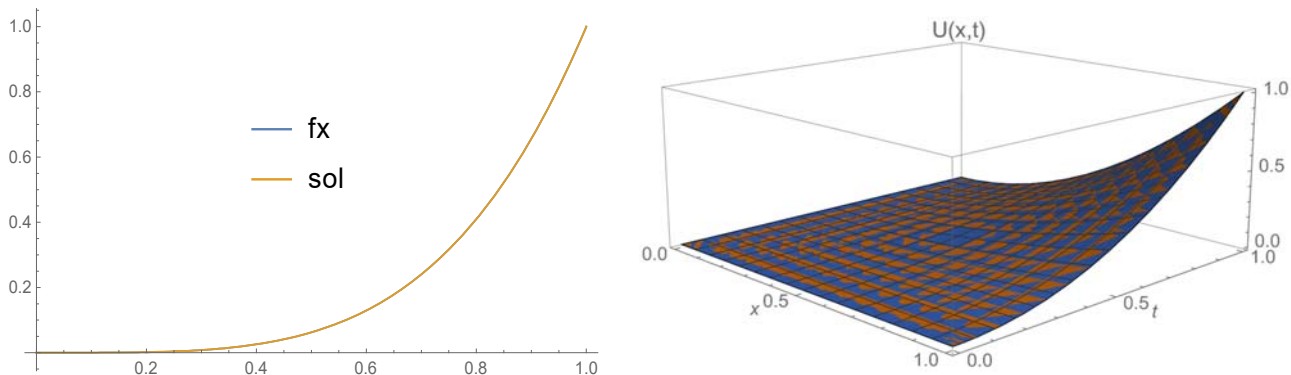

**Figure 3.** Graphs of the approximate $(fx)$ and the exact $(sol)$ results are presented on the left-hand side of the figure for $t = x$ and both solutions essentially overlap each other in one dimension. On the right-hand side, a 3D plot of exact and approximate solutions is provided over the intervals $t \in [0,1]$ and $x \in [0,1]$, which shows the accuracy of the numerical results because both graphs overlapped, showing no appreciable difference between them.

**Example 4.** *We shall now consider fourth example of the NPDE by substituting parameters $\alpha = 1$, $\beta = -1$, $\gamma = -1$, and $g(x,t) = -\dfrac{3\,x}{(2\,t+1)^2}$ into the general NPD Equation (1). So, the NPDE that we want to solve looks,*

$$\frac{d^2 y(x,t)}{dx^2} - y(x,t)\frac{d\,y(x,t)}{dx} - \frac{d\,y(x,t)}{dt} = -\frac{3\,x}{(2\,t+1)^2}. \tag{22}$$

The exact solution of Equation (22) is well known, $y_{exact} = \dfrac{3\,x}{(2\,t+1)}$. We are seeking a solution in the closed intervals $0 \le t \le 1$ and $0 \le x \le 1$ by applying initial condition at $t = 0$, $f(x) = y(x,0) = 3\,x$, $f'(x) = 3$, and $f''(x) = 0$. As we have done in

the previous examples, we would substitute the presumed solution Equation (4), into the NPD Equation (22) in order to convert it into matrix form. The projected solution to Equation (22) can be noted as $y(x, t) = \sum_{i=0}^{n} a_i(t) B_i(x) + 3x$, where in this expansion $a_i(t)$ is the *i*-th coefficient which depends on variable *t*. Putting this approximate solution into Equation (22) and multiplying both sides of Equation (22) with the product of B-polys $B_m(x) B_n(t)$, we can perform integration separately over both variables *t* and *x* in the intervals $t \in [0, T]$ and $x \in [0, R]$, respectively. Additional approximation to the coefficients $a_i(t) = \sum_{i=0}^{n} b_j^i B_j(t)$ can also be applied to obtain the following equation,

$$\sum_{i,j=0}^{n} b_j^i \left[ \langle B_i''(x) - 3x\, B_i'(x) - 3B_i(x)|B_m(x)\rangle\langle B_j(t)|B_n(t)\rangle - \sum_{k,l=0}^{n} b_l^k\, g_{ikm}' g_{jln} - \langle B_i(x)|B_m(x)\rangle\left\langle \dot{B}_j(t)|B_n(t)\right\rangle \right]$$
$$= \left\langle \left\langle \frac{-3x}{(2t+1)^2} + 9x|B_m(x)\right\rangle \middle| B_n(t)\right\rangle. \tag{23}$$

where, $g_{ikm}'$ and $g_{jln}$ are the nonlinear terms that exhibit nonlinearity via the unknown coefficients $b_l^k$. This process leads to an $(n+1)$ by $(n+1)$ order of equation $X B = W$, in terms of the unknown variables $B = \{b_1^1, b_2^1, b_3^1, \ldots, b_1^2, b_2^2, b_3^2, \ldots, \}$, where the operational matrix $X$ and the column matrix $W$ are,

$$X_{m.n} = \sum_{i,j=0}^{n} [\langle B_i''(x) - 3x\, B_i'(x) - 3B_i(x)|B_m(x) B_j(t)\rangle\langle |B_n(t)\rangle$$
$$- \sum_{k,l=0}^{n} b_l^k\, g_{ikm}' g_{jln} - \langle B_i(x)|B_m(x)\rangle\langle \dot{B}_j(t)|B_n(t)\rangle], \tag{24}$$
$$W_{m,n} = \left\langle \left\langle \frac{-3x}{(2t+1)^2} + 9x|B_m(x)\right\rangle|B_n(t)\right\rangle = \int_0^{T,R} \left( \frac{-3x}{(2t+1)^2} + 9x \right) B_m(x) B_n(t) dx\, dt.$$

It is tricky to choose the number of B-polys in both variable *x* and *t* for this example because the same degree set of B-polys can't be chosen in both variable *x* and *t*. We decided to choose $n = 1$ degree of B-polys in *x*-variable and $n = 13$-degree of B-polys in *t*-variable. This choice gave us a total of 28 B-polys basis set because *n* starts from 0. So, a large set of B-polys were used to calculate the solution of Equation (22). The composition of the exact solution shows that there is 1-degree of polynomials present in *x*, $\{1 - x, x\}$ and a series of the polynomials present in variable *t*. This is the reason we chose the approximate solution in this manner so that we minimize the error in the solution. The result converged after 15 iterations to the desired accuracy. Converged values of the 28 coefficients of the expansion for this example are also provided, $b_j^i = \{0, 0, 0, 0, 0, 0, 0, 0, 0, 0, 0, 0, 0, 0, 0, -0.46153, -0.76925, -1.00687, -1.19203, -1.34601, -1.47242, -1.58160, -1.67462, -1.75631, -1.82801, -1.89174, -1.94872,$ and $-1.99999\}$, which are the elements of matrix *B* needed to build the desired solution via Equation (24). Again, the accuracy and the quality of the results depend on the number of B-polys and the degree of polynomials used in the expansion of Equation (24). On the left side of Figure 4, when $t = x$ is substituted in both exact and the approximate solutions, we plotted a graph for comparison in linear dimension *x*. The graph shows the absolute error is of the order of $\sim 10^{-8}$ between the solutions. As the number of B-poly increases from second degree through 13-degree polynomials in variable *t*, the error between results (exact and approximate) was further decreased. This trend will be presented in the error analysis section. On the right-hand side of Figure 4, we also present a 3D plot of both solutions, which exhibits the accuracy of results in both variables *x* and *t*. The converged approximate solution of Example-4 is,

$$\begin{aligned} y(x, t) = &(3.0 - 5.99995\, t + 11.99753\, t^2 - 23.942006\, t^3 + 47.24151\, t^4 - 89.71297\, t^5 + 156.15497\, t^6 \\ &- 234.51855\, t^7 + 286.60518\, t^8 - 270.34520\, t^9 + 186.54281\, t^{10} - 87.84899\, t^{11} \\ &+ 25.09446\, t^{12} - 3.26879\, t^{13})\, x \end{aligned} \tag{25}$$

Graphs as well as excellent results of all examples indicate that the current technique works for solving a variety of linear [4] and nonlinear partial differential equations. We also present one B-polynomial used in $x$-variable, $\{x\}$, and 14 B-polynomials used in $t$-variable to obtain the results in Equation (25),

$$
\begin{aligned}
\{1 - 13t + 78t^2 \quad & -286t^3 + 715t^4 - 1287t^5 + 1716t^6 - 1716t^7 + 1287t^8 - 715t^9 + 286t^{10} - 78t^{11} + 13t^{12} \\
& - t^{13}, 13t - 156t^2 + 858t^3 - 2860t^4 + 6435t^5 - 10296t^6 + 12012t^7 - 10296t^8 + 6435t^9 \\
& - 2860t^{10} + 858t^{11} - 156t^{12} + 13t^{13}, 78t^2 - 858t^3 + 4290t^4 - 12870t^5 + 25740t^6 \\
& - 36036t^7 + 36036t^8 - 25740t^9 + 12870t^{10} - 4290t^{11} + 858t^{12} - 78t^{13}, 286t^3 - 2860t^4 \\
& + 12870t^5 - 34320t^6 + 60060t^7 - 72072t^8 + 60060t^9 - 34320t^{10} + 12870t^{11} - 2860t^{12} \\
& + 286t^{13}, 715t^4 - 6435t^5 + 25740t^6 - 60060t^7 + 90090t^8 - 90090t^9 + 60060t^{10} \\
& - 25740t^{11} + 6435t^{12} - 715t^{13}, 1287t^5 - 10296t^6 + 36036t^7 - 72072t^8 + 90090t^9 \\
& - 72072t^{10} + 36036t^{11} - 10296t^{12} + 1287t^{13}, 1716t^6 - 12012t^7 + 36036t^8 - 60060t^9 \\
& + 60060t^{10} - 36036t^{11} + 12012t^{12} - 1716t^{13}, 1716t^7 - 10296t^8 + 25740t^9 - 34320t^{10} \\
& + 25740t^{11} - 10296t^{12} + 1716t^{13}, 1287t^8 - 6435t^9 + 12870t^{10} - 12870t^{11} + 6435t^{12} \\
& - 1287t^{13}, 715t^9 - 2860t^{10} + 4290t^{11} - 2860t^{12} + 715t^{13}, 286t^{10} - 858t^{11} + 858t^{12} \\
& - 286t^{13}, 78t^{11} - 156t^{12} + 78t^{13}, 13t^{12} - 13t^{13}, t^{13}\}
\end{aligned}
$$

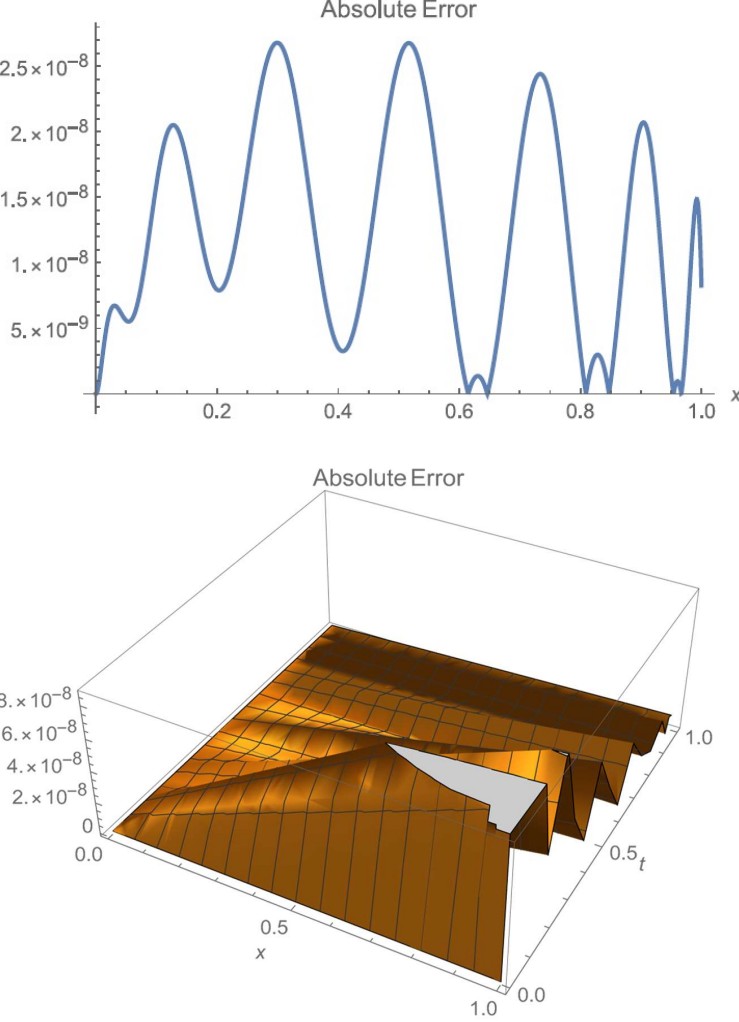

**Figure 4.** A plot of the absolute error between the exact and estimated solutions is depicted for $t = x$, on the left-hand side. The graph shows that the error is of the order of $10^{-8}$ in one dimension. A 3D graph of the absolute error for exact and estimated solutions is also given on the right-hand sde which shows the accuracy of the numerical results of the order of $10^{-8}$ over the intervals $t \in [0,1]$ and $x \in [0,1]$. To achieve this level of accuracy in the solution, a total of 28 B-polys basis set was involved which is a product of 2 B-polys in variable $x$ and 14 B-polys in variable $t$.

## 3. Error Analysis

It should be noted that the calculations based on B-polys are performed without a grid representation in the intervals. Numerical results depend on the number of B-polys and the degree of the polynomials chosen. In this section, we would like to show that as the number of B-polys are increased, the accuracy starts to improve. Both solutions, exact and approximate, start to overlap each other and the precision is achieved after a certain number of iterations of the nonlinear terms. We are going to present an error analysis for the approximate and exact solutions of the fourth example. The same error analysis can be performed on other NPD equations. As you have seen in Example 4, only a set of two B-polys of degree $n = 1$ was used in variable $x$, because by increasing the size of the B-poly basis set in variable $x$ did not help improve error. However, by increasing the set of B-polys in variable $t$ greatly helped improve the accuracy of the results of the NPDE. We can observe from the graphs that the absolute error decreases between the solutions as we steadily increase the set of B-polys basis in variable $t$. We are presenting two graphs for $n = 4$ and $n = 9$ degrees set of B-poly in variable $t$, while $n = 1$-degree of B-poly set was unchanged. These graphs are depicted in Figures 5 and 6 showing that how the error steadily decreases as we increase the number of B-polys from $n = 4$ through $n = 13$ degrees. The desired agreement between exact solutions started to converge as $n$ is increased for the inclusion of additional B-polys in variable $t$, see Figures 4–6. All the calculations are carried out without a grid on the intervals of integration. When $n = 4$ degrees of B-polys are used and calculations are iterated 15 times, the absolute error is found to be of the order of $10^{-3}$ which is shown in Figure 5. We have also presented the 3D graph of the absolute error in terms of two variables $(x,t)$ in Figure 6. In Figure 6, $n = 9$ degrees of polynomials were used to show that the error further decreased to the order $10^{-6}$. We have already shown a graph of the absolute error between the solutions, when $n = 13$ degrees of B-polys were chosen, see Figure 4. Clearly, the error is systematically decreased as the number of B-polys are increased in the calculations and each time results were iterated 15 times to achieve convergence of the solution. It indicates that the technique works to provide a converged solution that is comparable with the exact solution. The CPU time for the calculation significantly increased as we included a larger set of B-polys in the calculations.

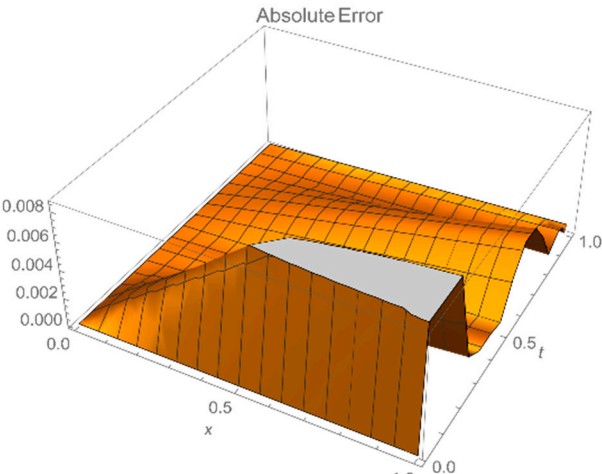

**Figure 5.** A 3D plot of the absolute error between approximate and exact solutions of Example 4 with $n = 4$-degree of B-polys is shown in two variables $(x,t)$. The approximate solution has not converged yet.

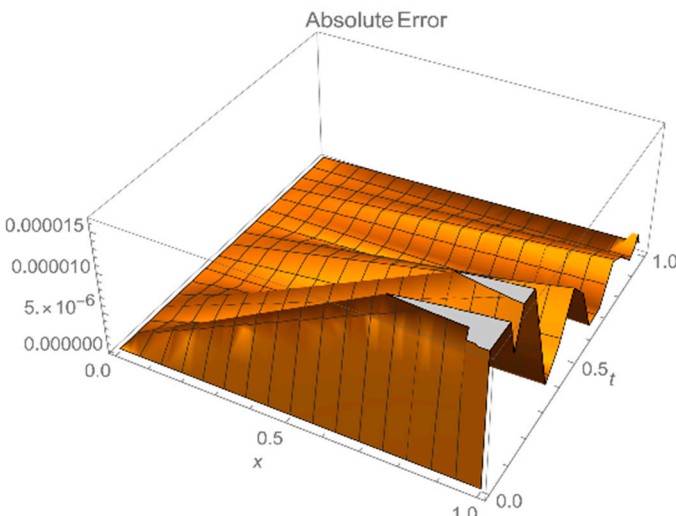

**Figure 6.** A 3D plot of the absolute error between approximate and exact solutions of Example 4 with $n = 9$-degree of B-polys is shown in both variables ($x,t$). The approximate solution has not converged as we need to increase the number of B-poly in variable $t$ for better accuracy in the solution.

## 4. Results and Discussion

In the present work, we have applied two-dimensional B-poly basis set technique to solve four examples of the NPD equations subjected to various initial conditions. Furthermore, A broader explanation of the 2D algorithm has been given to calculate precise results of the NPDE. We were approximating solutions employing the Galerkin method [17] in two variables ($x,t$) and the converged results have been found which are presented in Figures 1–4. Normally, these solutions converged after 10 to 15 iterations. For the first three examples, the approximate solutions were so precise that the solutions matched the exact solutions after ignoring the tiny contributions and a few iterations of nonlinear terms. With the increasing number of B-polys in the estimated solution, we have also observed that the accuracy of the solutions [1] increases. In the first two examples, we have used $n = 1$-degree set of polynomials in both variables and in the third example, $n = 2$-degree B-polys were used to obtain converged results. For the fourth example, only 1-degree polynomial in variable $x$ and 13-degree polynomials in variable $t$ were used to calculate the results. For Example 4, the B-poly basis set contained a total of 28 polynomials that were used to calculate the solution. We have also displayed graphs of the absolute error between the precise and the approximated results in Figures 1–4. In every case, the accuracy of the solutions has been different because different B-poly basis set have been used. When variable $t$ is set equal to $x$, the absolute errors between exact and approximate solutions have been compared. The accuracy was shown to be greater for the converged solutions as shown in the Figures 1–4. Our method worked very well for solving nonlinear and linear differential equations using an operational matrix scheme [13,18], as shown by the data and graphs presented in this paper. The Wolfram Mathematica symbolic program version 12 [19] was used to perform all analytic integrations and computations over the closed intervals for both variables ($x,t$).

Using B-poly basis sets, our method may display enormous possibilities for solving nonlinear and linear 2D problems in physics and other disciplines. Recently, many authors [20,21] have formed an operational matrix utilizing B-poly techniques to solve one-dimensional partial differential equations. We have successfully expanded this method to solving the two-dimensional NPDE. We also presented a detailed error analysis for the last example which shows how the error can be minimized as the number of B-polys are increased in the desired solution. To get a converged solution of the NPD Equation (22), only 15 iterations were used. The CPU time for executing Examples 1–3 took only about

3 min, while for Example 4, it took 10 min of CPU time as it involved higher degrees of B-polys and 15 iterations to converge.

In this article, we have shown an extended version of the technique [1] to solve NPD equations, based on the B-poly basis set that provided in some cases exact and suitable solutions of 2D nonlinear partial differential equations. This technique is superb for solving the problems related to a complex system of nonlinear partial differential equations where there are no known solutions that exist. We may employ this particular method in solving 2D linear and nonlinear partial differential equations as demonstrated in this paper.

**Author Contributions:** Conceptualization, M.I.B.; methodology, M.I.B.; software, M.I.B.; validation, M.I.B., N.D. and M.H.R.; formal analysis, M.I.B., M.H.R. and N.D.; investigation, M.I.B. and M.H.R.; resources, M.I.B.; data curation, M.I.B. and M.H.R.; writing—original draft preparation, M.I.B.; writing—review and editing, M.I.B., M.H.R. and N.D.; supervision, M.I.B.; project administration, M.I.B. and N.D. All authors have read and agreed to the published version of the manuscript.

**Funding:** This research received no external funding.

**Institutional Review Board Statement:** Not applicable.

**Informed Consent Statement:** Not applicable.

**Data Availability Statement:** Within the article of this study, the data that supports the findings are available.

**Acknowledgments:** We are indebted to the Departmental computational facilities which have been used to compute the results of this research paper.

**Conflicts of Interest:** The authors declare no conflict of interest.

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
