# Peer review of "Approximate Solutions of Nonlinear Partial Differential Equations Using B-Polynomial Bases"

_fractalfract, doi:10.3390/fractalfract5030106_

Round 1

Author Response

In my opinion, the article presents some interesting results. However, the authors should revise the manuscript to improve the paper. Here are some suggestions:

  1. The authors wrote: the currently employed technique provides a higher-order precision compared to the finite difference method. Please give a detailed comparison.

Response: We did not include reference to make things clearer.  For example, Kutluay et al [15] have presented a finite difference solution to the Burgers’ equation. It is shown that the accuracy of the solution depends on h2, where h is the size of the step chosen in the finite difference method.  This requires a very large number of grid points to accurately represent the solution in that region. In reference Bhatti & Bhatta [16], a combination of B-poly and 4th order Runge-Kutta method was applied to show that B-poly technique provided higher-order of accuracy (4 orders of magnitude better) in the solution of Burger equation.

2.The authors should specify the explicit forms of matrices X, B and W, and how to express equation (11) as XB = W.

Response: The equation (11) now has been expressed into matrices X, B and W which have been defined in equation (12).  The procedure shows how equation (11) transforms into equation X B = W. The matrices X and W are known and defined in eq. (12) and the matrix B has unknown coefficients which are determined by multiplying both sides of the equation with X-1 as mentioned in line from 151 through 157.

  1. How do you ensure the approximate solution y(x, t) = Pn i=0 ai(t)Bi(x) + f(x) expressed in the B-poly basis satisfy the initial condition y(x, 0) = f(x)? What are the B-poly basis in Example 2-4?

Response: Following the Galerkin method [17], the approximate solution of the partial differential equations is given by [16],

The initial condition at t=0, f(x) = y(x,0) is added to the solution given in eq. (4). The solution is substituted into the differential eq. to convert it into operational matrix X which is inverted to find the solution [16]. Before we find its inverse to be applied for determining unknown coefficients, boundary/initial conditions are applied to the operational matrix X and the right-hand side matrix W in the interval [a, b]. This is done by deleting the first row and the first column of the matrix X and the corresponding entry of the column matrix W to ensure solution vanishes at the origin.  Initial condition is satisfied by the approximate solution as explained in lines 143 and188.

The B-polys basis sets now have been included for each example considered in this paper. Further details of how to generate B-polys sets of various degrees, see ref. [1,3] and [16].

  1. There are some minor mistakes, such as, in line 40, “Differentiation” should be “Differential”, in formula (15), “Bi(t)” should be “Bi(x)”. The authors should check out the whole paper carefully. I propose a minor revision.

Response: Minor mistakes/typos have been corrected in the paper as pointed by the referee. All the minor proposed revisions have been implemented. Thanks.

Reviewer 2 Report

  1. There are too many words of partial differential equations in the key words of the paper. This does not seem to highlight the key content of the paper.
  2. The whole paper does not introduce what B-Polynomials are, nor does it introduce the theoretical basis that the solution of a partial differential equation can be expressed by B-Polynomials.
  3.  The paper also does not introduce the difficulty and importance of studying the equations (1) and (8).

Author Response

Comments and Suggestions for Authors

1.There are too many words of partial differential equations in the key words of the paper. This does not seem to highlight the key content of the paper.

Response: Too many words of partial differential equations in the key words have been reduced to highlight the key content of the paper. Thanks.

2.The whole paper does not introduce what B-Polynomials are, nor does it introduce the theoretical basis that the solution of a partial differential equation can be expressed by B-Polynomials.

Response: It is true that we did not give detail about the B-polys. The reason is that we wanted to avoid repetition. The specific details and properties of the B-polys are provided in our previous work in Refs. [1]- [3] and [13, 16]. However, now we have briefly included and defined B-polys in the introduction section, lines 34-37.

We have already introduced theoretical basis that the solution of partial differential equation can be expressed in terms of B-polynomials, see lines 6—68. Clearly mentioned that Galerkin Method [16] was employed to express the solution in terms of B-polys.

3.The paper also does not introduce the difficulty and importance of studying the equations (1) and (8).

Response: Two types of equations (1 & 8) are considered in this paper which are general second-order nonlinear partial differential equations. These equations are used for modeling real world physical phenomena in the fields of physics and mathematics. The equations have applications in various fields of science like fluid dynamics, heat conduction, elasticity, traffic control simulations and number theory [16] and references therein. Further, these equations can be converted into well-known Burgers’ equation if right hand side is set equal to zero and proper values of constants are chosen. The importance of these calculations is that sometimes they present exact solutions which can be used to judge algorithms as well as the accuracy of computational techniques, see ref. [13, 16].

These equations have nonlinear convective terms and second order diffusion terms which act opposite to each other. Furthermore, these terms allow to admit distorted wave profile as a solution which is difficult to solve numerically.

This has been added to the paper. Thanks.

Reviewer 3 Report

Please check carefully the manuscript and fix any typos/grammar issues.

Author Response

1.Please check carefully the manuscript and fix any typos/grammar issues.

Response: Manuscript has been checked and tried to fix typos/grammar mistakes. Thanks.

Reviewer 4 Report

This paper presents a multivariable technique to guesstimate the solution of nonlinear partial differential equations. The results obtained via the approach are accurate as evident from the absolute error. This review has the following concerns which require further clarification from the authors.

1. Is the presented technique applicable to fractional-order linear and nonlinear differential equations? Fractional-order differential equations are popular and strong tools for modeling, as also recognized by the authors in the introduction. Hence, additional examples on fractional-order linear and nonlinear differential equations should be provided to demonstrate the flexibility (if so) of the proposed formulation.

2. Quality of the figures and presentation should be improved.

3. In regards to the statement on the available approaches for obtaining solutions of fractional-order linear and nonlinear differential equations, additional works should be cited to better present the state-of-art:

a) A Ritz-based finite element method for a fractional-order boundary value problem of nonlocal elasticity. International Journal of Solids and Structures 202 (2020): 398-417.

b) Geometrically nonlinear response of a fractional-order nonlocal model of elasticity. International Journal of Non-Linear Mechanics 125 (2020): 103529.

c) Finite-element formulation of a nonlocal hereditary fractional-order Timoshenko beam. Journal of Engineering Mechanics 143.5 (2017): D4015001.

d) The finite element method for fractional non-local thermal energy transfer in non-homogeneous rigid conductors. Communications in Nonlinear Science and Numerical Simulation 29.1-3 (2015): 116-127.

Author Response

This paper presents a multivariable technique to guesstimate the solution of nonlinear partial differential equations. The results obtained via the approach are accurate as evident from the absolute error. This review has the following concerns which require further clarification from the authors.

  1. Is the presented technique applicable to fractional-order linear and nonlinear differential equations? Fractional-order differential equations are popular and strong tools for modeling, as also recognized by the authors in the introduction. Hence, additional examples on fractional-order linear and nonlinear differential equations should be provided to demonstrate the flexibility (if so) of the proposed formulation.

Response: There is a strong possibility that the current algorithm would work but that is not the subject in this paper. Our future-plan is to apply this technique to solve linear and nonlinear fractional-order differential equations as separate research work. Examples will be provided to demonstrate that the state-of-the-art formulism work in those cases. Indeed, fractional-order equations are becoming popular and strong tools for modeling. So, it provides strong motive to work in this area. Work is in progress. Thanks.

  1. Quality of the figures and presentation should be improved.

 Response: Improved quality of the figures has been included in the paper. Thanks.

  1. In regards to the statement on the available approaches for obtaining solutions of fractional-order linear and nonlinear differential equations, additional works should be cited to better present the state-of-art:

Response: We are glad the reviewer # 4 has provided 4 new references (a, b, c, and d) to be included concerning the fractional-order linear and nonlinear differential equations. As you may know we are systematically solving the multivariable differential equations. First, we applied the technique to solve ordinary and partial differential equations and published papers, see refs. [1,2, 3]. Second, we applied the extended technique to solve multidimensional linear differential equations, see ref. [13]. Third, the present work addresses how to solve nonlinear partial differential equations. Since the current work is only addressing the multivariable nonlinear differential using the B-poly basis set. We think these references may not be suitable to cite in the current paper under review. These references will be cited in upcoming publications regarding solving the multidimensional fractional-order differential equations which are going to require different approach to solve these types of equations. This group is already working in this direction.

a) A Ritz-based finite element method for a fractional-order boundary value problem of nonlocal elasticity. International Journal of Solids and Structures202 (2020): 398-417.

b) Geometrically nonlinear response of a fractional-order nonlocal model of elasticity. International Journal of Non-Linear Mechanics125 (2020): 103529.       c) Finite-element formulation of a nonlocal hereditary fractional-order Timoshenko beam. Journal of Engineering Mechanics5 (2017): D4015001.           d) The finite element method for fractional non-local thermal energy transfer in non-homogeneous rigid conductors. Communications in Nonlinear Science and Numerical Simulation1-3 (2015): 116-127.

Round 2

Reviewer 4 Report

Thanks for the clarifications. With regards to my previous comment 1: 

Since the authors do not treat fractional-order differential equations in this study, this reviewer wonders why do the authors then introduce fractional-order differential equations in the introduction, and then later refer to works that specifically treat fractional-order differential equations? In this regard, all references from [4]-[13] fractional-order differential equations?

I am referring to this abrupt statement in the introduction: "In the earlier years, various methods were employed to solve linear and nonlinear differential equations, including fractional-order differential equations [4]-[13]". 

Fractional-order differential equations are significantly different from integer-order differential equations, that as the authors clarified in this review, are their sole interest. So why digress the reader with such statements?

This manuscript is a resubmission of an earlier submission. The following is a list of the peer review reports and author responses from that submission.